# A New Parameterization of Photolysis Rates for Oxygenated Volatile Organic Compounds (OVOCs)

Yuwen Peng [1], Bin Yuan [1,*], Sihang Wang [1], Xin Song [1], Zhe Peng [1],

Wenjie Wang [2], Suxia Yang [1], Jipeng Qi [1], Xianjun He[1], Yibo Huangfu [1],

Xiao-Bing Li [1], Min Shao [1,*]

[1] College of Environment and Climate, Institute for Environmental and Climate Research, Guangdong-Hongkong- Macau Joint Laboratory of Collaborative Innovation for Environmental Quality, Jinan University, 51143, China

[2] Multiphase Chemistry Department, Max Planck Institute for Chemistry, Mainz 55128, Germany

*Correspondence to*: Bin Yuan (byuan@jnu.edu.cn) and Min Shao (mshao@pku.edu.cn)

**Abstract:**

Oxygenated volatile organic compounds (OVOCs) play a crucial role in atmospheric chemistry, significantly influencing radical production and VOC degradation through photolysis. However, current research on OVOC photolysis is limited by insufficient species coverage in the mechanisms and incomplete understanding from a species-specific perspective. In this study, the photolysis frequencies of 109 OVOCs were compiled into a comprehensive photolysis dataset. Based on their molecular structures, a parameterization for the photolysis frequencies of carbon- and nitrogen-containing OVOCs was developed. By establishing a relationship between species structure and photolysis frequency, this approach avoids the limitation of insufficient quantum yield data, enabling the estimation of photolysis rate constants for compounds lacking experimental measurements. Photolysis frequencies for the dataset species were successfully reproduced with 21 reference values and 10 adjustment coefficients. Using an automated program based on this method, photolysis rate constants for 3039 OVOCs were predicted, and the MCM v3.3.1 chemical mechanism was updated and expanded to include photolysis for 714 additional species. The introduction of the new photolysis mechanism has altered both the concentrations of photodegradable OVOCs and the relative proportions of their removal pathways. Non-HCHO OVOCs, particularly multifunctional species with carbonyl groups, contribute significantly to ROx radical production. At three different sites, non-HCHO OVOCs photolysis accounts for 25%-45% of ROx production, surpassing HCHO photolysis. The importance of oxidation products from aromatics and alkenes is highlighted, offering new insights into OVOCs photolysis from a species-specific perspective.

**Keywords:**

OVOCs photolysis, Parameterization, Box model, ROx radicals

## 1.Introduction

Oxygenated volatile organic compounds (OVOCs) are a subset of VOCs and encompass a range of compounds, including aldehydes, ketones, alcohols, acids, ethers, esters, and highly reactive species such as enals and enones. They significantly affect atmospheric chemistry and contribute to the formation of secondary pollution (Mellouki et al., 2015; Huang et al., 2020). OVOCs originated not only from the oxidation of hydrocarbons but also from primary emissions (Mcdonald et al., 2018). OVOCs are characterized by their diverse species, high reactivity, complex sources, and measurement difficulty. With the advancement of modern mass spectrometry technologies, the importance of OVOCs is gradually revealing, contributing 16%–51% to the total concentrations of VOCs in different environments (Liu et al., 2020; Li et al., 2020; Hui et al., 2018; Jing et al., 2020; Song et al., 2023). OVOCs are removed from the atmosphere through reactions with oxidants, wet or dry deposition, and photolysis (Mellouki et al., 2015). The photolysis rate influences the relative importance of removal pathways for photodegradable OVOCs. For some species, photolysis serves as the dominant sink and governs their atmospheric fates (Yuan et al., 2016; Eger et al., 2020).

Through photolysis, OVOCs can produce atmospheric radicals such as $RO_2$ and $HO_2$, which enhance atmospheric oxidation capacity and contribute to ozone formation (Tan et al., 2019a; Tan et al., 2019b; Kanaya et al., 2007; Griffith et al., 2016). The photolysis of OVOCs plays a significant role in the production of ROx radicals (ROx = $RO_2$ + $HO_2$+ OH) (See Figure 1 and Table S1) and can account for over 80% of the production of ROx (p(ROx)) during the later stages of winter ozone pollution (Edwards et al., 2014). OVOCs from different sources play varying roles in their contributions to atmospheric oxidation capacity. Primarily emitted OVOCs act as primary radical producers, capable of directly generating 1 to 2 radicals through photolysis, resulting in a net production of radicals. This net production provides a substantial supply of radicals for the initial oxidation processes in atmospheric chemistry (Mellouki et al., 2015). Meanwhile, OVOCs formed through secondary processes serve as

"photochemical amplifiers" accelerating the formation of secondary pollutants. OVOCs produced from oxidation may undergo photolysis to produce additional radicals, further contributing to formation of more OVOCs (Qu et al., 2021). This creates a positive feedback loop that has been recognized as a key factor in high winter ozone concentrations in multiple regions (Edwards et al., 2014; Li et al., 2021). Furthermore, as primary radical producers, OVOCs photolysis increases the production of ROx, while as photochemical amplifiers, they increase radical chain length. Thus, the photolysis of OVOCs effectively enhances atmospheric oxidation capacity.

However, several challenges remain that hinder our ability to better constrain the environmental effects of OVOCs photolysis. Firstly, current chemical mechanisms fail to fully account for recently reported photodegradable species, such as some multifunctional compounds (Newland et al., 2019; Tomas et al., 2021; Müller et al., 2014; Wang et al., 2023). Traditional mechanisms mainly focus on mono- or bi-functional, low-carbon species (See Section S1 and Figure S9 of *Supplementary Information*). Secondly, significant discrepancies exist in the reported photolysis frequencies of certain species (e.g., nitrophenol or benzaldehyde (Peng et al., 2023; Mellouki et al., 2015). These differences largely arise from varying estimates of quantum yield, which can lead to differences in photolysis rate constants by nearly an order of magnitude. Furthermore, many species remain unmeasured or lack sufficient reporting. Although the development of modern measurement techniques has greatly expanded the range of measurable species (Ye et al., 2021; Yuan et al., 2017), it is still prohibitively expensive in terms of time and cost to measure photodegradable species individually. Currently, the MPI-Mainz UV/VIS database includes absorption cross-section data for 310 organic compounds, but quantum yield information is only available for formaldehyde and acetaldehyde (Keller-Rudek et al., 2013). Similarly, the IUPAC database provides absorption cross-sections and photolysis product data for 39 compounds, yet more than half of these species lack recommended quantum yield values or assume a quantum yield of 1 across all wavelengths (IUPAC, 2012). The scarcity and uncertainty of quantum yield data remain significant limitations in deriving

accurate photolysis rate constants for OVOCs. Finally, due to the incomplete description of mechanisms, current evaluations for the contribution of OVOCs photolysis to atmospheric oxidation capacity primarily focus on simple species such as formaldehyde (Tan et al., 2019a; Young et al., 2012), or treat OVOCs as a whole (Liu et al., 2012; Yang et al., 2018).

Several parameterization approaches for OVOCs photolysis rate constants have been developed, including Master Chemical Mechanism (MCM) (Jenkin et al., 1997; Jenkin et al., 2003), the Generator for Explicit Chemistry and Kinetics of Organics in the Atmosphere (GECKO-A) (Aumont et al., 2005), and the SAPRC mechanism generation system (MechGen) (Carter et al., 2025). MCM v3.3.1 applies a core set of 20 reactions parameterized as functions of the solar zenith angle, with surrogate parameters for species with limited data, but sometimes assigns identical rates to structurally distinct compounds. GECKO-A focuses on three primary chromophores (carbonyls, hydroperoxides, nitrates) and employs detailed cross-section datasets for 54 species, yet lacks differentiation among conjugated systems or less common photodegradable functional groups. MechGen, developed specifically for the SAPRC mechanism, assigns overall quantum yields to representative species based on grouped reactivity. However, it assumes that quantum yields are wavelength-independent and provides only lumped quantum yields rather than explicit photolysis rate constants. Consequently, there is no comprehensive or easily accessible source for photolysis rate constants of a specific OVOC species, as the existing data are dispersed across various studies. These limitations highlight the need for a structure-specific approach that directly outputs photolysis rate constants and captures a wide range of functional group combinations.

In this study, we have integrated the most recently reported photolysis frequency data to obtain a dataset of OVOCs photolysis. From this dataset, we establish a relationship between the chemical structures of OVOCs and their photolysis rate constants, avoiding the need for inaccessible quantum yield data across many species. Based on this dataset, a new parameterization method was proposed to predict

photolysis frequencies based on molecular structures. Subsequently, these predicted photolysis frequencies were incorporated into chemical mechanisms and were evaluated using an observation-based box model to quantitatively assess the atmospheric impacts.

## 2 Method

### 2.1 Observation

Field measurements were conducted at three sites, representing both urban and regional environments in southern and northern China. The urban site in southern China was located at the Guangzhou Institute of Geochemistry (GIG), Chinese Academy of Sciences (23.1°N, 113.2°E) in the urban area of Guangzhou. Instruments were deployed approximately 25 meters above ground level, and measurements were taken during the fall of 2018 (September to November). The site was surrounded by residential areas and roadways (Wu et al., 2020; Wang et al., 2020; Yang et al., 2022). The urban site in northern China was in downtown Beijing, where a three-month field campaign was conducted from May to August of 2021. The site was located at the Institute of Atmospheric Physics (IAP), Chinese Academy of Sciences (23.15°N, 113.36°E), which is a large meteorological and environmental monitoring tower in the bustling city center (Li et al., 2025; Huangfu et al., 2024). Data collected from July 7 to July 31 at an approximate height of 5 m above the ground was used in this study. The regional site was the Guangdong Atmospheric Supersite (22.7°N, 112.9°E) situated in Heshan (HS), approximately 80 km southwest of Guangzhou and frequently influenced by anthropogenic emissions from the Guangzhou–Foshan megacity. Observations were conducted during the fall of 2019 (October-November), characterizing it as a representative regional receptor site in the Pearl River Delta (PRD) (Cai et al., 2024; Yang et al., 2022).

OVOCs were measured at all three sites using a high-resolution proton-transfer-reaction quadrupole interface time-of-flight mass spectrometer (PTR-QiToF-MS, Ionicon Analytik, Austria) (Wu et al., 2020; He et al., 2022). At the Guangzhou urban site and PRD regional site, non-methane hydrocarbons (NMHCs) were measured

hourly using a gas chromatograph equipped with a flame ionization detector and a mass spectrometer (GC-MS-FID, Wuhan Tianhong Co., Ltd, China) (Wang et al., 2022). NMHCs data from the Beijing urban site was unavailable so we estimated those unmeasured species according to proton-transfer-reaction time-of-flight mass spectrometer (PTR-ToF-MS) data, data from nearby stations, and OH reaction rate constants ($k_{OH}$) of individual species with OH radicals. Detailed estimation methods are provided in the Supporting Information of Li et al. (2025).

Details regarding the measurement of meteorological parameters, photolysis rates, and trace gases at the three sites, including instruments used and specific measurement methods, are provided in Section S2 of *Supplementary Information*. Further details on the instruments used at each site are provided in Wang et al. (2022) for the Guangzhou urban site, Yang et al. (2024) for the Beijing urban site, and Yang et al. (2022) for the PRD regional site. Temporal variations of typical photodegradable species at each site, along with reference parameters including Ox, NOx, O$_3$, temperature, relative humidity, and $j_{NO_2}$, are presented in Figure S1.

**2.2 Box Model Description**

A zero-dimensional box model, built on the Framework for 0-D Atmospheric Modeling (F0AM) v4.3 (Wolfe et al., 2016), was applied in this study to compare mechanisms and evaluate their atmospheric impacts. The chemical mechanisms included the MCM v3.3.1 (Jenkin et al., 2015; Bloss et al., 2005; Jenkin et al., 2003) and the photolysis mechanism proposed in this study. MCM has been widely used for modeling atmospheric radicals and secondary products (Aumont et al., 2005; Edwards et al., 2014; Chen et al., 2022). The model was constrained using 5-minute averaged relative humidity, ambient temperature, pressure, photolysis frequencies, and concentrations of directly measured trace gases and VOCs (listed in Table S2). Simulations were performed until steady-state conditions with a 3-day spin-up period to allow for the buildup of unmeasured intermediate species (Yang et al., 2022). To prevent the unrealistic accumulation of long-lived species, an empirically derived first-order physical dilution coefficient $k_{dil}$ was applied, with values of $1.2 \times 10^{-5}$ s$^{-1}$ for both

urban sites (equivalent to a 24-hour lifetime) and $3.5 \times 10^{-5}$ s$^{-1}$ for the PRD regional site
(equivalent to an 8-hour lifetime) (Yang et al., 2022). In addition to the MCM v3.3.1
mechanism, photolysis reaction of ClNO$_2$ has been included, frequency of which is
derived by reducing the $j_{NO_2}$ by a factor of 30 (Riedel et al., 2014).

In this study, we designed two model scenarios to explore atmospheric effects of

the newly proposed photolysis mechanism and to identify key species that significantly
contribute to the production of total radicals. Scenario 1 was designed to investigate the
impact of the new photolysis mechanism under traditional simulation settings. Among
all the OVOC species, only formaldehyde (HCHO) was constrained to its measured
concentrations. This setup allowed us to assess the impacts of the photolysis mechanism
on the atmospheric chemical simulations with the common practice, which facilitated
comparing our findings with other studies. However, a key limitation of Scenario 1 is
that it may not accurately simulate many organic intermediates. The inadequate
representation of these intermediates hindered a more detailed analysis of the key
species involved in atmospheric chemistry. To address this issue, Scenario 2 was
constrained by 20 additional non-HCHO OVOCs (listed in Table S4) based on PTR-
ToF-MS measurements and applied a dynamic allocation method to semi-quantitatively
estimate concentrations from PTR-ToF-MS, thereby constraining over 1300 OVOC
intermediates. A detailed description of the dynamic allocation method can be found in
Section S3 and Figure S10 of *Supplementary Information*. By incorporating a broader
range of OVOC constraints, Scenario 2 allowed for a more accurate representation of
atmospheric reactions and provided deeper insights into the role of OVOCs photolysis
in driving radical production. Aside from the differences in the constraints on OVOC
concentrations, both scenarios follow the same setup as described in the previous
section.
**2.3 Photolysis dataset for parameterization**

The photolysis frequencies can be calculated by numerical summation over

wavelength using Equation (1) (Calvert et al., 2002).

$$J = \int \delta_i \times \phi_i \times F_i \, d\lambda_i \qquad (1)$$

$\delta_i$, $\phi_i$, and $F_i$ stand for the absorption cross section, quantum yield, and spectral
actinic flux of the species $i$, respectively. The absorption cross-section and quantum
yield are typically obtained through laboratory measurements, while the actinic flux is
usually acquired through field measurements or models.
A dataset for OVOCs photolysis rate constants has been constructed for the
subsequent development of parameterization. The data within the dataset are collected
from databases like MPI-Mainz UV/VIS (Keller-Rudek et al., 2013) and IUPAC
(Mellouki et al., 2021), as well as from textbooks (Calvert et al., 2011) and other
relevant literature (See the *"Species Dataset"* sheet in the supplementary file for details).
The tropospheric ultraviolet and visible (TUV) radiation model (version 5.3)
(Madronich and Flocke, 1999; Lantz et al., 1996) was used to provide a spectral actinic
flux under a clear-sky condition. Before inclusion into the dataset, all data were
converted to the ratio of photolysis frequencies of OVOCs species under overhead sun
condition ($j_{OVOC,overhead}$) to those for the reference species ($j_{NO_2,overhead}$), which is
referred as the relative photolysis frequencies $j_{rel}$ ($= j_{OVOC,overhead}/j_{NO_2,overhead}$).
The relative photolysis frequencies can be conveniently applied to different
environments by conversion with measured $j_{NO_2}$. The data inclusion process is depicted
in Section S4 and Figure S11 of *Supplementary Information*.
The photolysis dataset is divided into a reference group and a comparison group.
In the current photolysis dataset, the reference group encompasses a total of 195
photolysis rate constants data entries for 109 OVOCs species, including 5 pairs of
isomers. The comparison group includes 50 data entries covering 38 species from 8
reports. The reference group primarily consists of data from smog chamber
measurements and peer-reviewed recommended data, which will be used for the
construction of parameterization schemes. The data in the comparison group mainly
consists of the results measured at a single wavelength, results obtained by measuring
absorption cross-sections but estimating quantum yields, and estimates for entire
categories of substances, such as the single value provided by Treves and Rudich (2003)
for hydroxy nitrates containing 3 to 6 carbon atoms. Additionally, results from quantum
chemical calculations are also included in the comparison group. Since it mainly
includes data with larger uncertainties than the reference group, the comparison group
is only used for comparison and not for the construction of the parameterization.

For all OVOCs species in the dataset, the simplified molecular input line entry

system (SMILES) format is used to describe their molecular structures. SMILES is a
specification that uses ASCII strings to explicitly describe molecular structures. It is not
only capable of covering all chemical formulas, but also highly readable, easy to
understand, and has been adopted by many leading international chemical databases
(Weininger, 1988; Wang et al., 2018).

## 3 Results and discussion

### 3.1 Development of a structural-based parameterization method

An analysis of the relationship between the relative photolysis rate constant ($j_{rel}$)

and species characteristics revealed no significant correlation between $j_{rel}$ and the
number of oxygen atoms in the molecules. However, some aromatics, nitrates,
aldehydes, and hydroxy carbonyls show a slight increase in $j_{rel}$ with rising carbon
oxidation states ($\overline{OS_c}$), calculated following the method of Kroll et al. (2011), while
other species display no significant trends. Interestingly, $j_{rel}$ tend to increase with the
number of functional groups in the molecules, except for N-nitrosamines and alkyl
nitrites (Figure S2). Although no clear correlation was observed between $j_{rel}$ and the
number of carbon atoms, compounds with similar functional groups tend to cluster
within a specific range of $j_{rel}$ (Figure 2). When $j_{rel}$ of the species in the same
compound class are averaged, distinct differences emerge between functional group
types, suggesting that the parameterization should focus on the types and quantities of
functional groups present in the species.

The photolysis data for OVOCs collected from multiple literature sources were

classified based on their functional groups. Compounds within the same family,
particularly those with similar functional groups, generally exhibit analogous
absorption cross-sections in the UV-visible spectrum (see Figure S3). This property
serves as an essential theoretical basis for structure-based estimation. During oxidation
processes, when new functional groups are added to existing molecular structures,
conjugation may occur between different chromophores. Such interaction cannot be
simply treated as the additive effects of two individual chromophores, making it
necessary to discuss different conjugated systems categorically.
A comparison of data from different literature sources revealed consistent
photolysis rate constants for most species, with notable differences observed for certain
compounds, such as benzaldehyde, pyruvic acid, peracetic acid, and 2-butenedial
(Figure S4). Based on the 195 entries in the OVOCs photolysis dataset, the reported
values for the same compound from different literature sources were first averaged,
obtaining the averaged data for 109 OVOCs ("*Averaged Dataset*" sheet in the
supplementary file). The averaged $j_{rel}$ for 109 compounds were then classified into 23
categories according to their functional group types. These $j_{rel}$ form a dataset for the
development of a more accurate parameterization.
Based on the summarized photolysis frequencies, the reference values for 21
classes of compounds were calculated to provide guidance for predicting the photolysis
frequencies of the compounds in the same classes containing more carbon atoms.
Taking the example of n-aldehydes, one of the most extensively studied groups, their
absorption band appears in the UVB region (around 280 nm) due to a weak $n \rightarrow \pi^*$
electronic transition, overlapping partially with the tropospheric radiation spectrum
(>290 nm) (Calvert et al., 2011). The absorption cross-sections of n-aldehydes
containing 1 to 7 carbon atoms are shown in Figure S11. For formaldehyde (HCHO),
the carbonyl carbon is attached with two hydrogen atoms, and the stretching vibration
of the C=O double bond in the $\pi^*$ excited state is particularly prominent in the
absorption band, resulting in significant fluctuations in its absorption cross-section. As
the carbon chain length increases, the stretching vibration of the C=O bond in the $\pi^*$
excited state becomes less pronounced, and the absorption bands nearly overlap
(Calvert et al., 2011). Although the absorption of n-heptanal (n-$C_6H_{13}CHO$) is slightly
higher than that of n-pentanal, it is reasonable to consider that, for n-aldehydes, the
absorption cross-sections and photolysis rate constants become similar once the carbon
number reaches 3 or higher. This assumption is consistent with a previous study (Tadić
et al., 2001). Therefore, in calculating the reference photolysis rate constants for the
aldehyde carbonyl group (-CHO), the average photolysis rate constants of aldehydes
with 3 to 7 carbons are used as the reference value ($0.18\% \pm 0.053\% \, jNO_2$) for
aldehydes with 8 or more carbons. In addition to n-aldehydes, photolysis reference
values were calculated for other 20 classes of species, including substituted aldehydes
(e.g., nitrobenzaldehydes, aromatic aldehydes, unsaturated aldehydes, and dialdehydes),
nitrogen-containing compounds (e.g., N-nitrosamines, nitrophenols, nitroaromatics,
alkyl nitrites, carbonyl nitrates, and peroxy nitrates), ketones (e.g., diketones, cyclic
ketones, and unsaturated ketones), as well as other species such as dienedials, keto acids,
hydroperoxides, and peroxy acids (see Figure 3). The reference values for
corresponding species with 1 to 20 carbons are tabulated for clarity (See the *"Reference*
*Value Table"* sheet in the supplementary file). By establishing a relationship between
the photolysis rate constants and carbon number, the reference photolysis rate constants
for higher-carbon compounds are estimated. The calculation methods for these
functional groups are similar to those of aldehydes, with details provided in Section S5
and Figure S12-S30 of *Supplementary Information*. The parameterization is currently
applicable only to compounds composed of carbon (C), hydrogen (H), oxygen (O), and
nitrogen (N). Functional groups involving sulfur, halogens, or other elements are not
involved in this study, due to limited availability of data for these compounds. For
similar reason, the current parameterization does not include epoxy compounds or
anhydrides either.
In addition to the reference values, adjustment coefficients were introduced to
reflect the influence of different molecular structures more accurately on photolysis rate
constants. For example, when 2 methyl groups substitute both α-carbons of 3-pentanone
(forming 2,4-dimethyl-3-pentanone), the absorption significantly increases, likely due
to a redshift caused by the inductive effect of alkyl groups, resulting in an enhancement
of $0.36\% \, jNO_2$ for the photolysis frequencies (due to the lack of photolysis data for 3-
pentanone, 2-pentanone is used as a surrogate here). By identifying such relationships

and evaluating them against the corresponding reference values for their respective carbon numbers, a total of ten adjustment coefficients have been introduced in the current parameterization. The specific basis and calculation process for other parameters can be found in Section S6 and Figure S31-S33 in the *Supplementary Information*. The introduction of adjustment coefficients enhances the applicability and accuracy of the parameterization for estimating photolysis rate constants of compounds with diverse molecular structures.

Finally, to enable the estimation for a vast number of OVOCs, we developed a program to automate structure recognition and photolysis frequency estimation, as shown in Figure 4 for the workflow of the program. The program utilizes the SMILES as the input format to identify the carbon count and characterize both the types and quantities of functional groups. Then, the corresponding reference value and adjustment coefficients from the pre-established dataset are retrieved to calculate and output the predicted photolysis frequencies of the OVOC in the form of $j_{rel}$. This approach effectively reproduces the $j_{rel}$ of 104 compounds from the reference group of the dataset (Figure 5a). The main deviations arise from underestimations for nitro-naphthalene, 2-acetylbenzaldehyde, and 2-hexenal, as well as overestimation for hydroxy butanone, highlighting areas for future optimization. Compared to the measured data in the dataset, 81% of the species fall within a ±50% deviation range, and 98% within a factor of 2 (Figure 5a). The strong agreement between measured and predicted values ($R^2$ = 0.92) indicates the reliable performance of this proposed parameterization method (Figure 5a).

For the 50 species in the comparison group, the estimated results from this study align well with $j_{rel}$ of approximately half of the species (Figure S5). Taking nitrocatechols as an example, their $j_{rel}$ were estimated based on those of structurally similar nitrophenols, as both share aromatic rings substituted with OH and $NO_2$ groups. The parameterization effectively reproduces the $j_{rel}$ of four types of nitrocatechols measured by Roman et al. (2022) using chamber experiments, further demonstrating the reliable performance of the method. Notable deviations mainly originate from two

studies. Wang et al. (2023) reported significantly lower $j_{rel}$ for three classes of
monoterpene-derived organic nitrates, while Liu et al. (2018) estimated the $j_{rel}$ of α-
hydroperoxycarbonyls to be only about 7% of our estimated value. These differences
can be explained by the substantial uncertainties in the comparison dataset, which was
derived from eight independent studies that employed a variety of methodologies, such
as quantum chemical calculations, single-wavelength measurements, and indirect
estimates based on absorption cross-sections combined with assumed quantum yields.
In addition, the current method does not incorporate structure-specific adjustments for
certain compound classes, which may also contribute to the observed discrepancies.
**3.2 Comparison to Master Chemical Mechanism v3.3.1**
As a detailed mechanism, MCM includes 5809 species and intermediates, among
which 2327 are classified as photodegradable. Our estimates of $j_{rel}$ align with MCM
within ±50% for 61% of the evaluated species (Figure 5b). Among all the
photodegradable species in the MCM, two species (CH3SOOOH and ETOMENO3)
are identified as non-photodegradable, as the method used is not applicable to sulfur-
containing species and ETOMENO3 lack photoreactive chromophores. Consequently,
no photolysis rate constants were generated for them. The differences in photolysis rate
constants primarily arise from differences in the classification and estimation of
functional groups for multifunctional compounds (Figure 5b). The parameterization
method produces higher values for species such as aromatic aldehydes, multi-carbonyl
compounds, nitrate esters, and hydroxyl nitrate esters, while it yields lower values for
species including carbonyl hydroperoxides, unsaturated diketones, hydroxy nitrates,
and hydroxycarboxylic acids when compared to MCM. The detailed estimation results
for 2327 photodegradable species can be found in the "*Prediction-MCM2327*" sheet in
the supplementary file. An updated MCM v3.3.1 mechanism that can be directly used
in the F0AM box model is available in the file named
"*MCMv331_OVOCshv_updated_Mech*" on the FigShare website (Peng, 2025).
In addition to these 2327 photodegradable species identified in MCM, this study
further identifies 714 additional photodegradable species from the 5809 species in
MCM based on their structural characteristics. The 714 newly identified species are
primarily characterized by their diverse chromophore, including carbonyl groups,
nitrates (e.g., $ONO_2$, $C(=O)ONO_2$, $C(=O)OONO_2$), and hydroxyl groups (OH). A
notable proportion (38%) of these species contain ring structures, suggesting their
origin from aromatic hydrocarbon, terpenes, or other polycyclic compounds.
Additionally, 46% of the newly identified species contain two or more chromophores,
further emphasizing the need for more comprehensive evaluation of multifunctional
species. The photolysis rate constants of these photolysis reactions were determined
using the structure-based parameterization, while the reaction products were inferred
based on existing patterns of known products, producing 1 to 2 radicals. Details of the
newly coupled photolysis module with MCM v3.3.1 are provided in the "*714 Newly*
*Add Species*" and "*Additional OVOCs Mech*" sheet in the supplementary file. A
supplementary mechanism that can be directly used in the F0AM box model is available
in the file named "*MCMv331_OVOCshv_updated_Mech* " on the FigShare website
(Peng, 2025). Details on surrogate products for unknown photolysis reactions can be
found in Section S7 of *Supplementary Information*. The modifications to photolysis
reactions, including photolysis rate constants and reaction products are incorporated
into the standard MCM v3.3.1 as a newly updated chemical mechanism
The application of the new photolysis mechanism has influenced both the
concentration of photodegradable OVOCs and the relative contributions of their
removal pathways. Compared to the standard MCM mechanism, a 50% or greater
increase in the concentration of OVOCs was observed for 9% of photodegradable
species at Guangzhou urban site, and for 16% at PRD regional site (Figure 6 (a, b)).
Some species exhibited an increase in concentration even when photolysis was
enhanced, which was observed for 9.2% of species at Guangzhou urban site, and for
11% at PRD regional site. This observation may be attributed to the positive feedback
between the radicals generated by the enhanced photolysis of these OVOCs and the
oxidation of their precursors, leading to the formation of additional photodegradable
products. This effect is consistent with the findings of Qu et al. (2021), where OVOC
photolysis contribute to the amplification of radical cycling in the atmosphere.

The changes in concentrations of OVOCs reflect variations of their atmospheric

budget. Here, the ratio of $j_{OVOC}/(k_{OH} \times [OH])$ is used to represent the relative
importance of two pathways for each OVOC species: photolysis and reactions with OH
radicals (Figure 6 (c, d)). The ratio is calculated by dividing the photolysis removal rate
by the OH reaction removal rate for each species, and the campaign average is then
computed to obtain the mean ratio for each species. After adopting the new photolysis
scheme, the relative importance of photolysis increased for over half of the
photodegradable OVOCs species at both sites. For the campaign-averaged
$j_{OVOC}/(k_{OH} \times [OH])$, the new mechanism shows a 25% increase (from 1.2 to 1.5) at
Guangzhou urban site and a 21% increase (from 1.7 to 2.1) at PRD regional site. From
the perspective of the number of carbonyl functional group in each species, species
containing zero to one carbonyl group is removed from the atmosphere mainly via
reaction with OH radical rather than photolysis. For species containing two or more
carbonyls, photolysis dominates their atmospheric removal. The averaged
$j_{OVOC}/(k_{OH} \times [OH])$ for compounds with 2 or more carbonyls is 4.4 at the Guangzhou
urban site and 6.1 at the PRD regional site, indicating that photolysis is a more
important removal pathway than reaction with OH radicals for multi-carbonyl
compounds. At the Guangzhou urban site, the new mechanism increased the
$j_{OVOC}/(k_{OH} \times [OH])$ value by 34% for species with 0-1 carbonyls and by 188% for
species with more than three carbonyls, while the ratio decreased by 27% for species
with two carbonyls. Similar changes were observed at the PRD regional site. This
suggests that previous mechanisms may overestimate the photolysis of dicarbonyl
compounds, while underestimation for photolysis of multi-carbonyl compounds. Late-
generation VOC oxidation products, especially those with multiple carbonyl groups,
warrant further investigation into their role in photolysis as part of their atmospheric
removal pathway.

The updated mechanism not only influences OVOCs themselves but also affects

the production of ROx radicals in our simulations. A significant contribution from non-

HCHO OVOC photolysis to ROx radical production was observed at both sites, with an enhancement in p(ROx) upon application of the updated photolysis mechanism. After implementing the updated mechanism, the averaged daytime contribution of non-HCHO OVOCs photolysis to p(ROx) increased from 36% to 40% at the Guangzhou urban site (Figure 7). In contrast, the contribution from HCHO photolysis remained relatively low at 13%. Similarly, at the PRD regional site, the averaged daytime contribution of non-HCHO OVOCs photolysis increased from 44% to 47%, which is 2.6 times that of HCHO photolysis (Figure 7).  As a result, the total daytime p(ROx) reached 2.9 ppb $h^{-1}$ at the Guangzhou urban site and 3.7 ppb at the PRD regional site. Notably, at noon when ROx production peaked, p(ROx) showed an increase of 7.6% at the Guangzhou urban site and 8.8% at the PRD regional site. These findings highlight the dominant role of non-HCHO OVOCs photolysis in driving ROx radical production, surpassing the contribution from formaldehyde photolysis.

Under the influence of the new photolysis mechanism, more than 2000 OVOC species collectively contributed to changes in p(ROx) levels. The implementation of this new mechanism led to an increased contribution to p(ROx) from non-carbonyl-containing OVOCs (e.g., nitrophenols), unsaturated mono-carbonyl species (e.g., methacrolein, MACR), and saturated tri-carbonyl compounds (e.g., 3,4-dioxopentanal) at both sites (Figure 7 Comparison of the modeled contributions of different OVOC types to total p(ROx) at urban and regional sites under Scenario 1. Blue bars represent the urban site, while yellow bars represent the regional site. Error bars (this study only) show the range from using the maximum and minimum estimated photolysis rate constants in the simulations.

). The greatest absolute increase in p(ROx) was observed for unsaturated mono-carbonyl compounds at the PRD regional site, followed by non-carbonyl-containing OVOCs, with mean increases of 2.4 ppb $d^{-1}$ and 1.0 ppb $d^{-1}$, respectively. In contrast, the contribution of unsaturated di-carbonyl compounds (e.g., 2-methylbutenedial) to p(ROx) decreased at both sites. The impact of the new photolysis mechanism was more pronounced at the PRD regional site, likely due to the higher fractions of OVOCs in downwind areas (Liang et al., 2022).

As radical production is increased using the new photolysis mechanism, ozone production is also enhanced. The enhancement of ozone production rate p(O₃) at noon was relatively modest, with increases of 5.3% at the urban site and 3.9% at the regional

site. Similarly, peroxyacetyl nitrate (PAN), a typical photochemical product, exhibited
minor sensitivity to the new mechanism, with noon concentrations increasing by 2.2%–
6.9% at both sites. This may result from modifications to the photolysis rate constants
of a wide range of OVOCs in the new mechanism, where the opposing effects of
increased and decreased rates effectively counterbalanced each other, masking the
impact of OVOCs photolysis on secondary species formation.
**3.3 Identifying key non-HCHO species contributing to p(ROx)**
Only HCHO concentrations were constrained in Scenario 1 of model simulation,
which may bear substantial biases within simulated concentrations of other
photodegradable OVOCs and hence influence identification of key species that
significantly contribute to p(ROx). For example, the new mechanism incorporated the
photolysis of nitrophenol, partially alleviating its overestimation, yet the simulated
concentrations remained significantly higher than observations (see Figure S7). This
residual overestimation may be attributed to enhanced secondary formation driven by
accelerated ROx cycling, as well as the absence of heterogeneous loss processes such
as particle-phase reactions and deposition. To identify key species among numerous
photodegradable OVOCs, Scenario 2 imposes extensive constraints on intermediate
products to approximate a "quasi-realistic" VOC composition. Under these conditions,
the impacts of the new photolysis mechanism on p(ROx) are investigated across three
different environments. OVOCs contributed over 50% of VOCs at all sites, with the
highest OVOC contribution at the PRD regional site (56%).
After constraining intermediate species in the model, non-HCHO OVOCs still
show a significant contribution to p(ROx). At the urban sites, non-HCHO OVOCs
contributed 27% (Guangzhou) and 25% (Beijing) to p(ROx), while at the PRD regional
site, this contribution was higher at 45%, with an average p(ROx) of 16 ppb d$^{-1}$ (Figure
9 (a,c,e)). Despite similar $j_{NO_2}$ values at the PRD regional site and the Guangzhou urban
site (Figure S8), the PRD regional site showed a significantly higher p(ROx), indicating
that higher OVOC concentrations and more aged air masses are key drivers of radical
production in the regional environment. This contrast suggests that under these

conditions, the composition and concentration of OVOCs play a more crucial role in radical production than photolysis intensity.

Further analysis of the relative fractions reveals regional differences in the contribution of non-HCHO OVOC photolysis to p(ROx) throughout the day. At the PRD regional site, the contribution of non-HCHO OVOC increased with solar radiation, peaking at 46% at noon, then stabilizing before declining after sunset (Figure 9 (d)). This peak corresponds to maximum radiation, and the afternoon stability suggests replenishment by the oxidation of transported VOCs from upwind megacities. In contrast, the contribution at both urban sites peaked at 35% at 7 a.m. and then gradually declined to 25%–30% before sharply dropping prior to sunset (Figure 9 (b, f)). The morning peak is likely due to primary emissions or photolysis of overnight-accumulated OVOCs, while lower daytime values reflect limited photodegradable OVOCs in fresher air. In summary, the contribution of non-HCHO OVOC photolysis to p(ROx) exhibited a unimodal diurnal pattern at the PRD regional site, while a morning peak was observed at both urban sites.

An analysis of the contribution of non-HCHO OVOC photolysis to p(ROx) was conducted based on the structure of OVOCs (Figure 10 (a, c, e)). Both urban sites exhibited similar trends, with OVOCs containing a single carbonyl group dominating their contribution to p(ROx). At the Beijing urban site, these mono-carbonyl OVOCs accounted for nearly half of the non-HCHO OVOCs, most of which were saturated compounds. Mono-carbonyls contributed approximately 13% to the total p(ROx) at the Guangzhou urban site and 14% at the Beijing urban site, given that non-HCHO OVOCs contributed around 27% and 25%, respectively. Di-carbonyl OVOCs contributed second-most at both urban sites, mostly from saturated compounds, while compounds with 3 to 5 carbonyl groups contributed 9%–12%. In contrast, di-carbonyl compounds were the primary contributor at the PRD regional site, with an increased contribution of unsaturated mono-carbonyls. Compounds with 3 to 5 carbonyl groups contributed 17% at the PRD regional site, significantly higher than at urban sites, highlighting the role of more oxidized and multifunctional species in regional environments.

Compounds without carbonyl groups, such as nitrophenol and organic nitrates, contributed minimally (5-8%) across all sites, likely due to lower concentrations or slower photolysis rates. Generally, mono-carbonyl OVOCs dominate p(ROx) at urban sites, while multiple carbonyl OVOCs are more important at the regional site.

From an individual species perspective, the top ten non-HCHO OVOCs accounted for more than half of the total non-HCHO OVOC contribution to p(ROx) at all sites (Figure 10 (b, d, f)). At urban sites, the contributions are primarily driven by OVOCs containing 1-2 carbonyl groups, contributing 50% at the Guangzhou urban site and 45% at the Beijing urban site. At the PRD regional site, contributions from di-carbonyl and multi-carbonyl (3-4 carbonyl groups) compounds increased significantly. The top ten non-HCHO OVOCs across all sites mainly included oxidation products from alkenes, ring-retaining products and multi-carbonyl ring-opening products from aromatic oxidation, as well as aldehydes. Notably, biogenic VOC oxidation products such as those from isoprene and β-pinene (e.g., NOPINONE) were prominent at both urban sites, highlighting the potential importance of BVOCs in OVOC formation in urban areas. Although the individual contributions of the remaining OVOCs (over 2000 species) outside the top ten were small, their cumulative impact remained notable.

## 4 Conclusion

In this study, a structure-based parameterization for OVOCs photolysis rate constants has been developed, together with an OVOCs photolysis mechanism supplementary module based on the MCM v3.3.1 mechanism. The proposed method effectively reproduces photolysis rate constants for 104 compounds using 21 reference values and 10 adjustment coefficients. Additionally, it identifies 714 photodegradable species not considered photolyzable in MCM v3.3.1. The integration of the new photolysis mechanism leads to a 13% increase in p(ROx) at both Guangzhou urban site and PRD regional sites, contributing 27% and 56% to daily p(ROx), respectively. For multi-carbonyl compounds, photolysis contributes more to their atmospheric removal than reactions with OH radicals. After innovatively using the model results to allocate mass spectrometry data and constrain more intermediate species, the model still reveals

significant contributions of non-HCHO OVOC photolysis to p(ROx). Non-HCHO OVOC photolysis contributes 25%–27% to p(ROx) at both urban sites, and 45% at the PRD regional site, surpassing the contribution of HCHO photolysis. The diurnal variations in the fraction of p(ROx) contributed by non-HCHO OVOC photolysis highlight the importance of primary OVOC emissions at urban sites and the oxidation of transported VOCs at the PRD regional site. The top 10 species contribute over half of the non-HCHO OVOC contribution to p(ROx), while the remaining more than 2000 species collectively make a noteworthy contribution. Photolysis of multifunctional compounds produced from the oxidation of alkenes and aromatics plays a major role in radical production.

By establishing a relationship between species structure and photolysis frequencies, this study circumvents the limitation of insufficient quantum yield data, enabling the derivation of photolysis mechanisms for compounds whose photolysis frequencies have not been previously measured. The estimated rate constants, combined with species concentrations from model simulations, allow for the identification of key species that significantly contribute to radical production in different environments. This approach not only provides a species-specific evaluation of OVOCs photolysis, but also offers valuable insights for future laboratory experiments, field observations, and the optimization of chemical mechanisms. Nevertheless, some species remain insufficiently characterized in terms of their photolysis rate constants or mechanisms. For example, the photolysis rate constant of nitrophenol varies notably among different studies. The photolysis behavior of pinene oxidation products has been scarcely examined (Wang et al., 2023), and in this study, their mechanisms were surrogated by using those of cyclo-ketones. Additionally, several compounds with strong photolysis potential, such as alkyl nitrites and N-nitrosamines, are not yet included in the MCM mechanism and require further investigation. Research on multifunctional OVOCs, particularly multi-carbonyl compounds, is still limited. Future experimental efforts are expected to generate more comprehensive data, thereby refining the existing OVOCs photolysis mechanisms, and

ultimately enhancing our understanding of atmospheric chemistry.

## Data and code availability

The observational data and parameterization tool-kit used in this study are available from corresponding authors upon request (byuan@jnu.edu.cn).

## Author contributions

YWP, BY and MS designed the research. SHW, XS, WJW, SXY, JPQ, XJH, YBHF, XBL contributed to field campaign and data collection. YWP performed the data analysis and parameterization of photolysis data. YWP, BY, and MS prepared the manuscript. All the authors reviewed the manuscript.

## Competing interests

The authors declare that they have no known competing financial interests or personal relationships that could have appeared to influence the work reported in this paper.

## Acknowledgements

This work was supported by the National Natural Science Foundation of China (grant No. 42275103, 42121004) and National Key Research and Development Program of China (grant No. 2023YFC3706200).

## Appendix A. Supplementary Information

Supporting information to this article can be found at …

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

 **Figures**

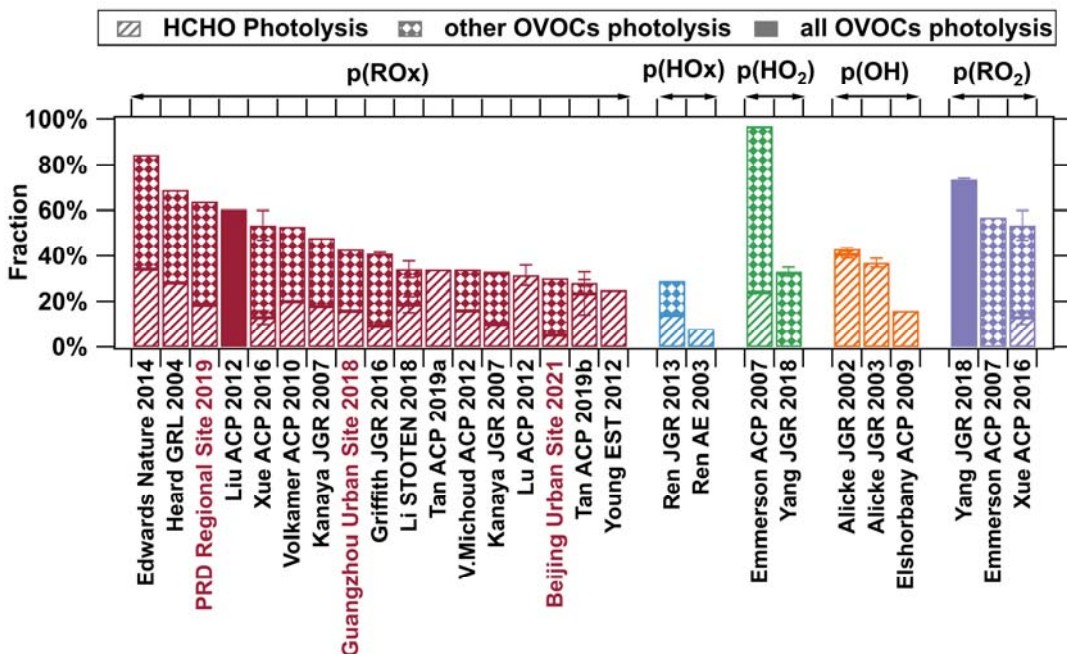

**Figure 1** Contributions of OVOC photolysis to the production rates of radicals in different environments. ROx represents the sum of $HO_2$, $RO_2$, and OH, while HOx is the sum of $HO_2$ and OH. The contributions of OVOCs photolysis at the three sites determined in this study are shown in red. Detailed data, locations, season and year of the measurements, and references are provided in Table S1.

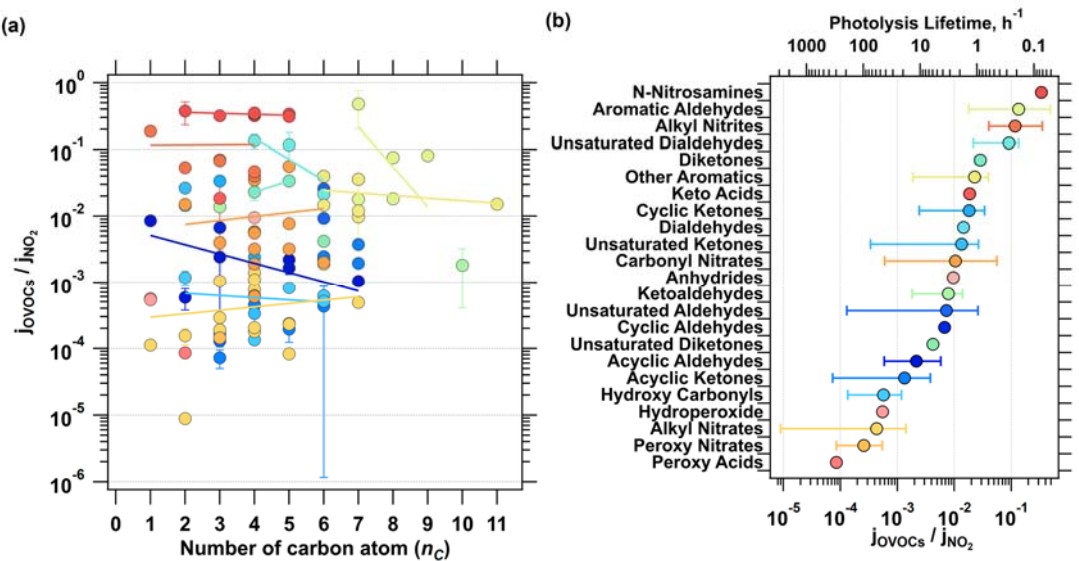

**Figure 2** Relationship between the relative photolysis rate constants of OVOCs and the number of carbon atoms (a), and the range of relative photolysis rate constants for different categories of OVOCs after averaging with each class (b). The top axis of subplot (b) indicates the corresponding lifetimes against photolysis under overhead sun conditions, based on the $j_{rel}$, as a reference.

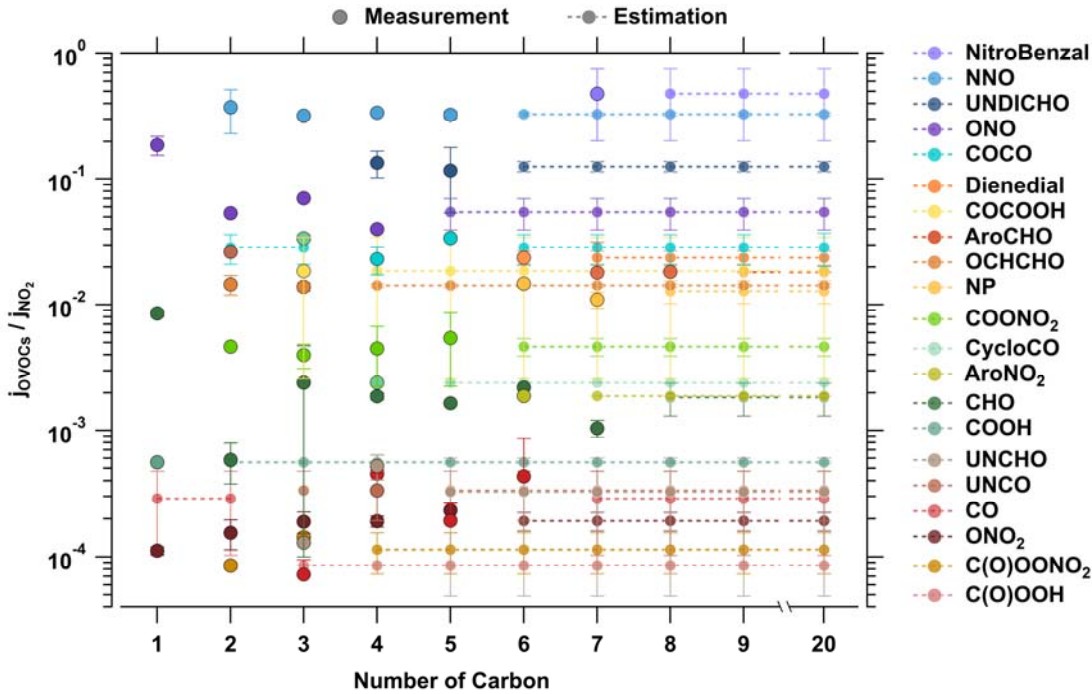

**Figure 3** Variation of photolysis reference values of 21 different classes of OVOCs with the numbers of carbons in the molecules. The scatter points in the figure represent measured values from the dataset, while the dashed lines and the points connected by them indicate predicted values for the higher carbon-containing species.

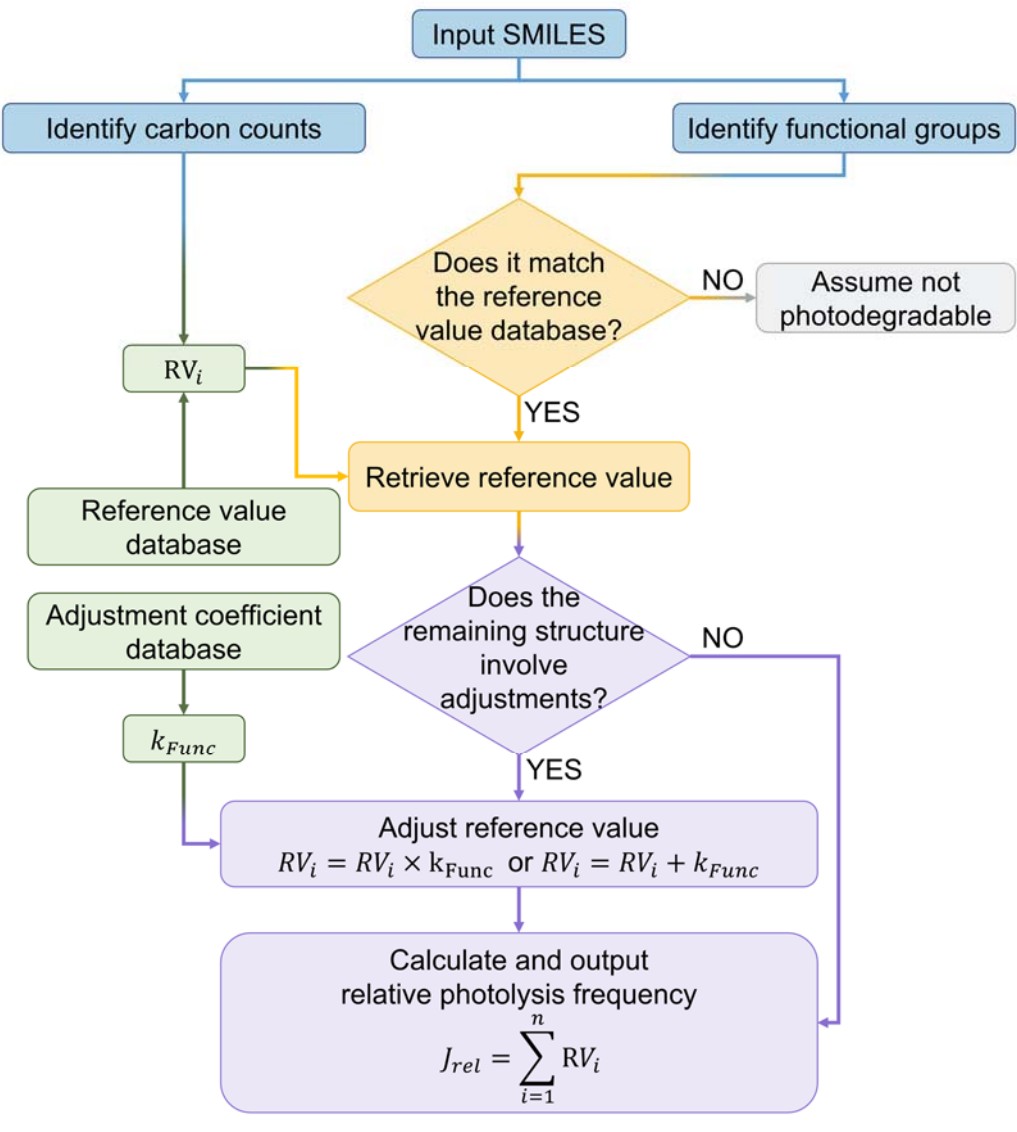

**Figure 4** Flowchart of structural-based photolysis frequency parameterization.

910

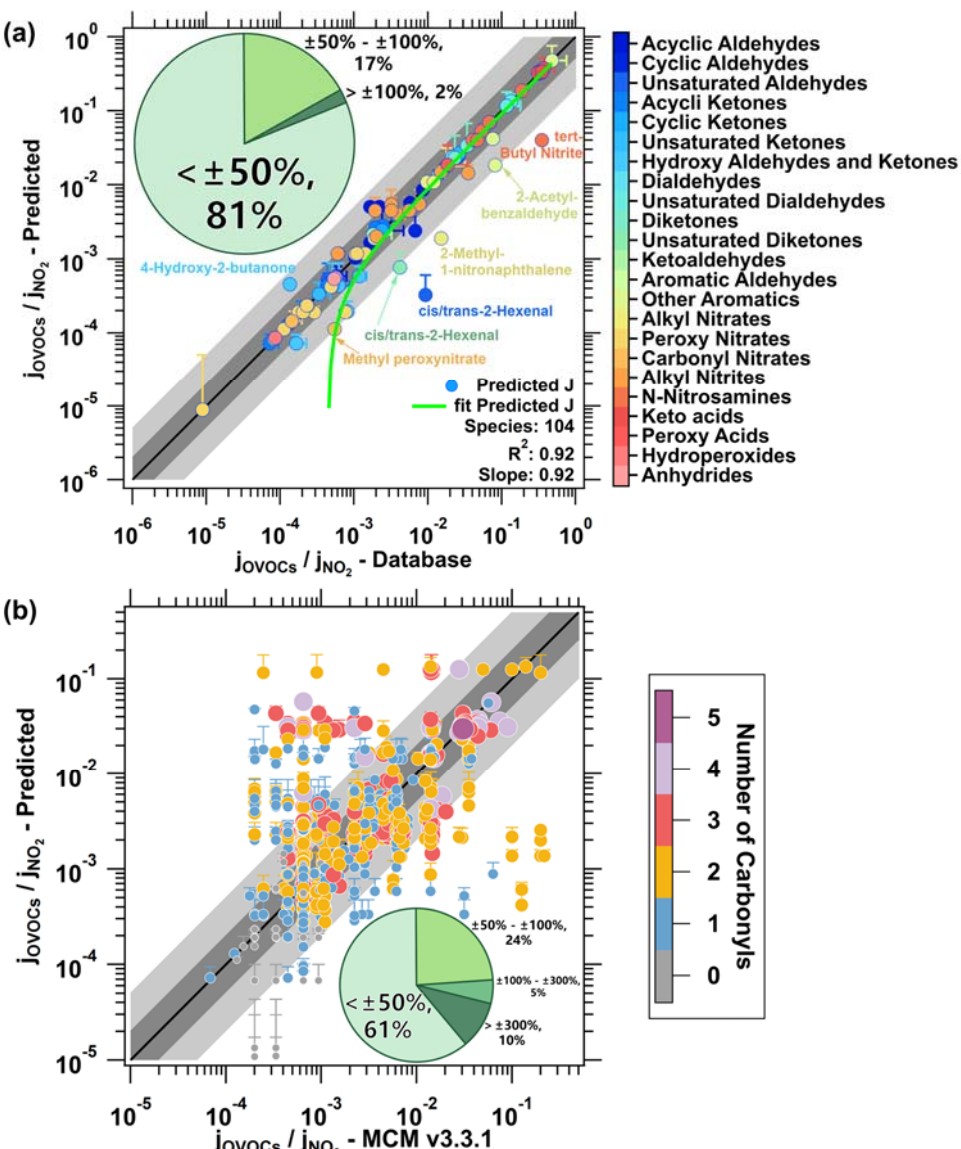

**Figure 5** Comparison of predicted $j_{rel}$ in this study and the measured $j_{rel}$ in the reference group (a) or the $j_{rel}$ in the MCM v3.3.1 mechanism. Each point corresponds to a specific compound, with species showing larger deviations labeled for clarity. The green curve in subplot (a) represents the fitted prediction. Different colors and sizes of the points in subplot (b) represent the number of carbonyl groups in each species. The black line indicates the 1:1 agreement and the darker gray band represents deviation of a factor of 2, while the lighter gray band extends to deviation for a factor of 5. The subplot pie chart illustrates the distribution of the relative deviation, calculated as $(predicted - measured)/measured$.

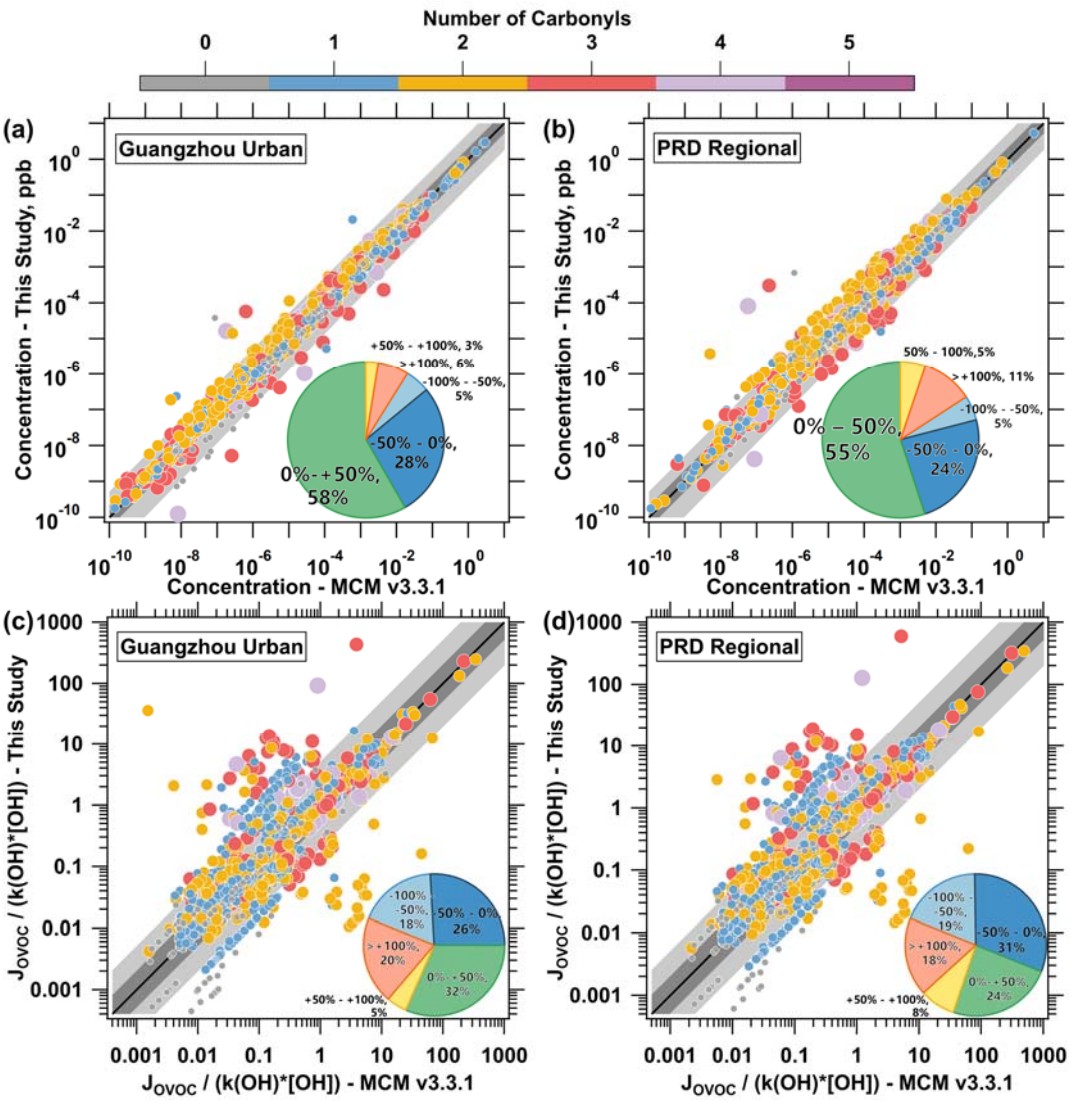

922

**Figure 6** Comparison of concentration of photodegradable species (a, b), and the ratio of $j_{OVOC}/(k_{OH} \times [OH])$ (c, d) in the MCM V3.3.1 mechanism and predictions from this study in Scenario 1. Each point represents a specific compound. Different colors and sizes of the points represent the number of carbonyl groups in each species. The black line indicates the 1:1 agreement and the darker gray band represents deviation of a factor of 2, while the lighter gray band extends to deviation for a factor of 5. The subplot pie chart illustrates the distribution of the relative deviation, calculated as $(This\ Study - MCM)/MCM$ .

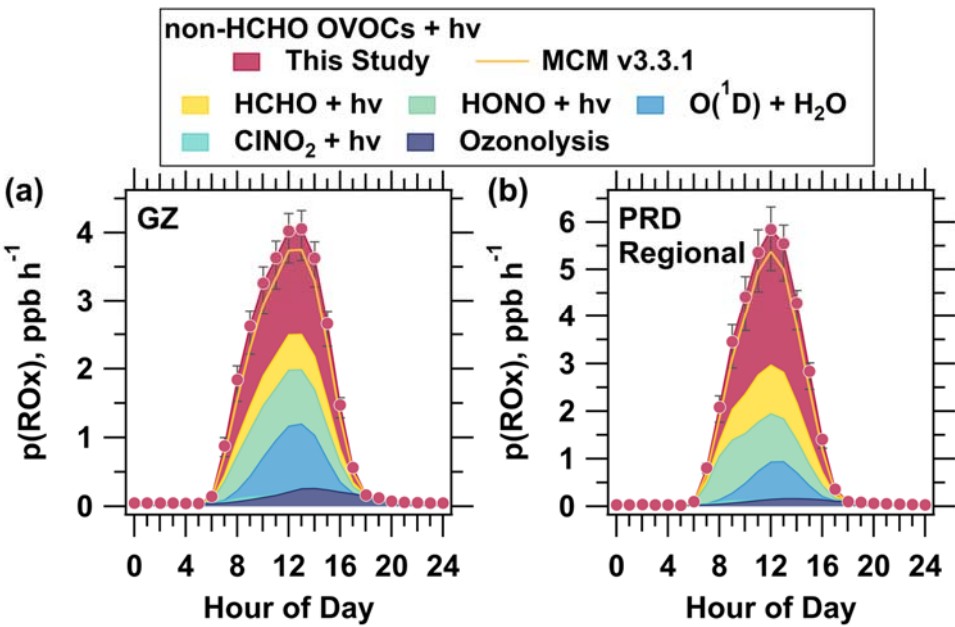

931

**Figure 7** Comparison of the modeled contributions of different OVOC types to total p(ROx) at urban and regional sites under Scenario 1. Blue bars represent the urban site, while yellow bars represent the regional site. Error bars (this study only) show the range from using the maximum and minimum estimated photolysis rate constants in the simulations.

937

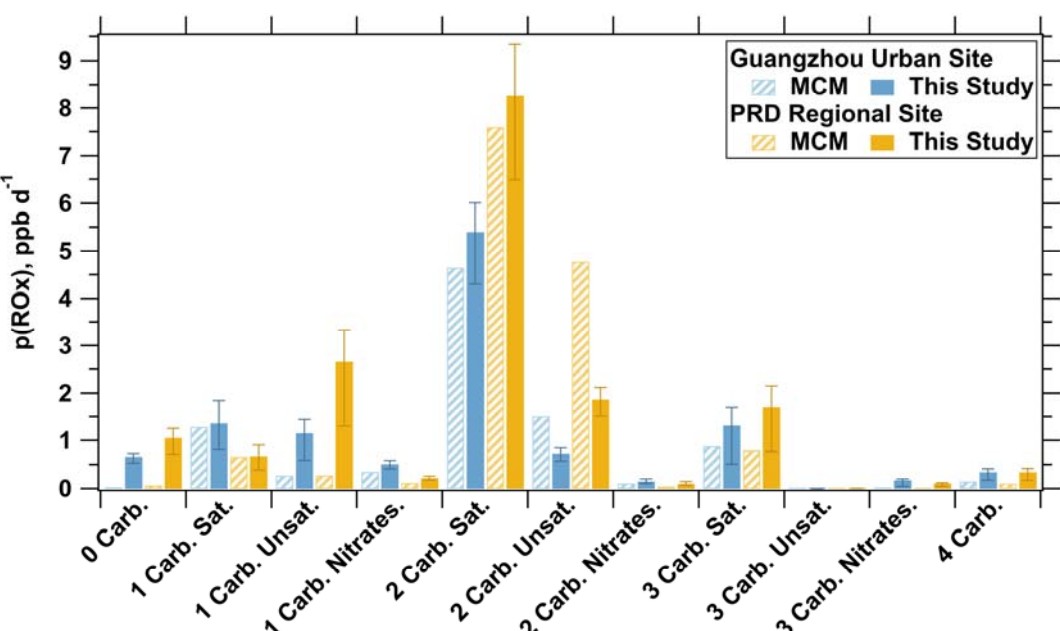

938

**Figure 8** Comparison of the modeled contributions of different OVOC types to total p(ROx) at urban and regional sites under Scenario 1. Blue bars represent the urban site, while yellow bars represent the regional site. Error bars (this study only) show the range from using the maximum and minimum estimated photolysis rate constants in the simulations.

944

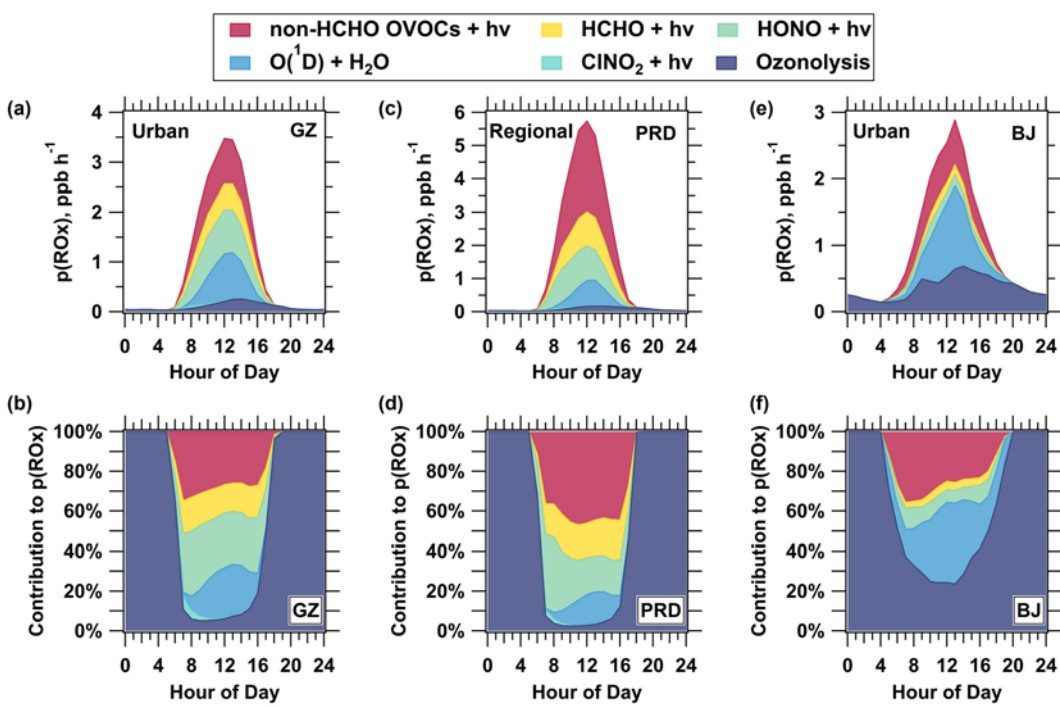

945

**Figure 9** Source composition and pathway contributions of total p(ROx) at three sites under scenario 2. The first row (a, c, e) represents the p(ROx) source composition at the Guangzhou urban site (GZ, a), PRD regional site (PRD, c), and Beijing urban site (BJ, e), respectively. The second row (b, d, f) shows the contributions of different pathways to p(ROx) at the corresponding sites.

951

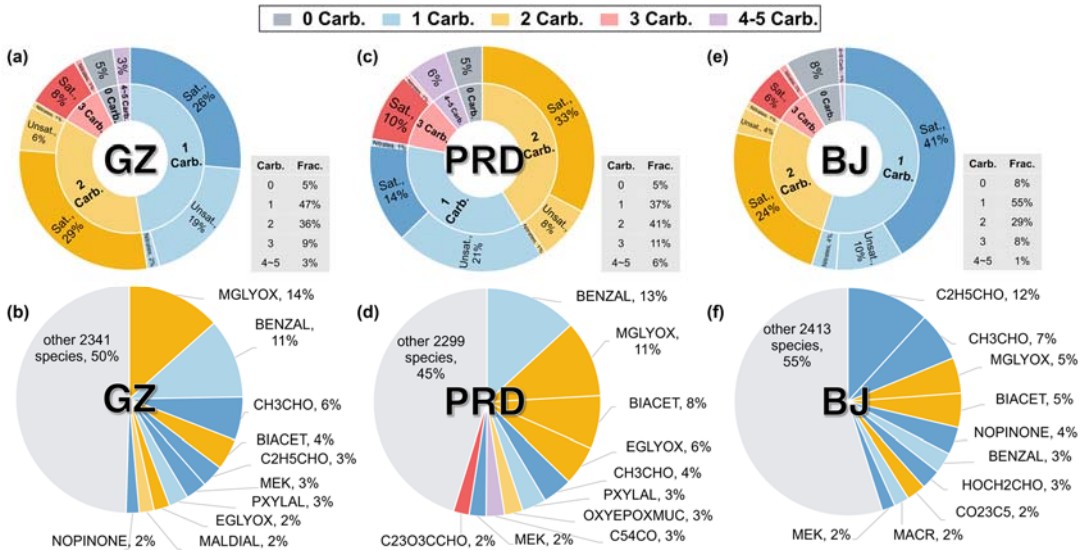

952

**Figure 10** Contribution of OVOCs with different carbonyl counts to p(ROx) and the top 10 contributing species in Scenario 2. The first row (a, c, e) shows the fraction of OVOCs with different numbers of carbonyl groups, where **Sat.** stands for saturated, **Unsat.** for unsaturated, and **Nitrates** represents organic nitrates. Similar colors indicate species with the same number of carbonyl groups. The gray table at the bottom right of the sunburst chart displays the total fraction for each category of carbonyl groups. The second row (b, d, f) presents the MCM names and specific contributions of the top 10 contributing species. Results for the Guangzhou urban site are shown in (a, b), for the PRD regional site in (c, d), and for the Beijing urban site in (e, f).