# Peer review of "A New Parameterization of Photolysis Rates for 1 Oxygenated Volatile Organic Compounds (OVOCs) 2 Yuwen Peng1, Bin Yuan 1,\*, Sihang Wang1, Xin Song1, Zhe Peng1, 3 Wenjie Wang 2, Suxia Yang 1, Jipeng Qi 1"

_EGUsphere, 2025_

## Referee Comment (RC2)

**A New Parameterization of Photolysis Rates for Oxygenated Volatile Organic Compounds (OVOCs)**

**General comments**

This study presents a novel and well-executed parameterization method for estimating the photolysis rate constants of oxygenated volatile organic compounds (OVOCs) based on molecular structure. By constructing a photolysis module that complements the MCM v3.3.1 mechanism, the authors successfully establish a structure–reactivity relationship that overcomes the longstanding limitation of insufficient quantum yield data. This allows for the estimation of photolysis rate constants for a wide range of compounds lacking experimental measurements. Furthermore, by integrating the updated photolysis mechanism, the study highlights the significant contribution of non-formaldehyde OVOCs to the daily average concentration of ROx radicals. Overall, this work introduces a creative and highly valuable approach for addressing a critical gap in atmospheric chemistry modeling. Some technical suggestions for improving the manuscript are provided below.

**Specific comments**

(1) Line 28, Please note that 'photolysis rate' is different from 'photolysis rate constant.' It would be more appropriate to use 'photolysis rate constant' here and in other relevant parts of the manuscript.

(2) Line 155, there is a Chinese-style period that needs to be replaced.

(3) The Supplementary Information includes numerous tables and figures; however, several figures are not referenced in the main text. It is recommended that the authors incorporate appropriate citations and briefly discuss the relevance of these figures to enhance clarity and coherence.

---

## Author Comment (AC1)

**Reviewer #1**

This work presented a novel method to estimate photolysis rates for OVOCs and uses an observational based box model to evaluate the contribution of OVOCs photolysis to the production rate of total radicals. However, the manuscript requires the inclusion of additional information to help readers understand and potentially further apply the proposed method. I recommend that the authors provide additional explanations and make minor revision to clarify their methods prior to publication. The authors may consider the following suggestions:

Reply: We thank the reviewer for the thoughtful comments and for recognizing the novelty of our method. In response, we have added additional explanations and comparisons to better illustrate the rationale and applicability of our approach. Please find the response to individual comments below.

1. Line 104: The authors only provided an overview of the three methods without summarizing and comparing their respective advantages and disadvantages. It is recommended to include a comparative discussion to highlight the strengths and weaknesses of the existing approaches, thereby better motivating the new method proposed in this study.

Reply: Thank you for the valuable suggestion. We have restructured the paragraph to provide a clearer and more informative comparison of MCM v3.3.1, GECKO-A, and MechGen. In addition to describing their estimation strategies, we now explicitly highlight their respective limitations. This comparison better motivates the need for our new structure-based parameterization method, which aims to overcome these challenges and provide more accurate and accessible photolysis rate estimations.

The fourth paragraph in introduction (line 109-123) are modified to:

*MCM v3.3.1 applies a core set of 20 reactions parameterized as functions of the solar zenith angle, with surrogate parameters for species with limited data, but sometimes assigns identical rates to structurally distinct compounds. GECKO-A focuses on three primary chromophores (carbonyls, hydroperoxides, nitrates) and employs detailed cross-section datasets for 54 species, yet lacks differentiation among conjugated*

*systems or less common photodegradable functional groups. MechGen, developed specifically for the SAPRC mechanism, assigns overall quantum yields to representative species based on grouped reactivity. However, it assumes that quantum yields are wavelength-independent and provides only lumped quantum yields rather than explicit photolysis rate constants. Consequently, there is no comprehensive or easily accessible source for photolysis rate constant of a specific OVOC species, as the existing data are dispersed across various studies. These limitations highlight the need for a structure-specific approach that directly outputs photolysis rate constants and captures a wide range of functional group combinations.*

2.    The newly proposed photolysis mechanism was only compared with the results from MCM. Could the authors also compare it with other chemical mechanisms? If not, please explain the reason.

Reply: The newly proposed photolysis mechanism is a structure-based parameterization designed to estimate photolysis rate constants for individual species. Among major atmospheric chemical mechanisms, most adopt lumped species representations to reduce chemical complexity in 3D models. In contrast, the MCM provides species-specific representations with explicit molecular identities; for example, toluene (TOLUENE), o-xylene (OXYL), and m-xylene (MXYL) are treated as distinct species. In lumped mechanisms such as SAPRC07, all aromatic hydrocarbons with OH reaction rates lower than $2 \times 10^4$ ppm$^{-1}$ min$^{-1}$ are grouped into ARO1, which includes structurally diverse compounds, some of which are photolyzed in the troposphere while others are not. As a result, our method cannot be directly compared with such lumped species. In future work, we may explore estimation of photolysis rates for representative structures within lumped groups and averaging them to provide reference values for broader inter-mechanism comparisons.

3.    The method developed in this study is based on the ratio $j_{(OVOC)}/j_{(NO2)}$. Given the wavelength-dependent absorption cross-sections of different species and functional groups, would using $j_{(NO2)}$ as a reference introduce potential biases under varying

environmental conditions?

Reply: This is an excellent point. As noted in Section 2.3, the $j_{rel}$ used in this study were calculated under overhead sun conditions and can be as upper-limit estimates. Indeed, due to differences in absorption wavelength distributions, the ratio of OVOC to $NO_2$ photolysis rate constants may vary under different solar zenith angles (SZA). This variation mainly depends on the spectral overlap between the absorption cross sections of the target OVOC and $NO_2$. For example, the overlap of the $Cl_2$ absorption spectrum with that of $NO_2$ is quite strong, resulting in a nearly constant $j_{rel}$ across different SZA values. In contrast, there is minimal overlap between the active wavelength regions of acetone and $NO_2$; acetone absorbs more strongly at shorter wavelengths, leading to a ~43% decrease in its $j_{rel}$ as the SZA increases from 0° to 90° (Calvert et al., 2011). While this does not typically affect the order of magnitude of $j_{rel}$, it can introduce bias. Future versions of the parameterization could incorporate an SZA-dependent correction factor to better account for this effect, further improving the accuracy and applicability of the approach. A clarification has been added to the manuscript at line 227:

*Before inclusion into the dataset, all data were converted to the ratio of photolysis frequencies of OVOCs species under overhead sun condition ($j_{OVOC,overhead}$) to those for the reference species ($j_{NO_2,overhead}$), which is referred as the relative photolysis frequencies $j_{rel}$ ($= j_{OVOC,overhead}/j_{NO_2,overhead}$).*

4. Line 469: After incorporating the new photolysis mechanism, the model simulations still significantly overestimate the observed values. What might be the possible reasons for this discrepancy?

Reply: The overestimation after incorporating the new photolysis mechanism may be attributed to two factors. First, the inclusion of additional photolysis pathways can accelerate ROx radical cycling, which enhances the oxidation of precursor VOCs and promotes the secondary formation of OVOCs such as nitrophenols, potentially creating a self-amplifying effect. Second, the current model does not explicitly account for heterogeneous loss processes on particles. Since nitrophenols are known

to undergo deposition or transformation on particulate matter, the omission of these sinks may result in an overestimation of their gas-phase concentrations. An explanation of this issue has been added to Section 3.3 of the manuscript (line 489):

*This residual overestimation may be attributed to enhanced secondary formation driven by accelerated ROx cycling, as well as the absence of heterogeneous loss processes such as particle-phase reactions and deposition.*

5.  Figure 7, I would suggest to add sub panels to compare the non-HCHO OVOC photolysis between this work and MCM calculation with error bars to show the potential difference. Similarly, error bars should be added in Figure 8.

Reply: Thank you for your thoughtful suggestion. Regarding Figure 7, we acknowledge the value of using subpanels for comparison. However, after careful consideration, we decided to retain the integrated panel format in the main figure, as only the "non-HCHO OVOCs + hv" pathway differs between the two mechanisms, while the other five pathways are identical. Splitting the figure into sub-panels may introduce unnecessary complexity without significantly enhancing clarity. To address this point more specifically, we have added a supplementary figure (Figure S6) that presents a focused comparison of the "non-HCHO OVOCs + hv" pathway contributions from this study and the MCM, which we believe better highlights the targeted difference.

Following the advice, we have added error bars to Figure 7 and Figure 8 to better illustrate the potential variability in our results. The uncertainty ranges were derived from simulations conducted using the upper and lower limits of the estimated photolysis rate constants. As the MCM mechanism provides fixed photolysis parameters without associated uncertainties, no error bars are shown for MCM results in the figure.

[Figure]

Figure 1 Comparison of the modeled contributions of different OVOC types to total p(ROx) at urban and regional sites under Scenario 1. Blue bars represent the urban site, while yellow bars represent the regional site. Error bars (this study only) show the range from using the maximum and minimum estimated photolysis rate constants in the simulations.

[Figure]

Figure 2 Comparison of the modeled contributions of different OVOC types to total p(ROx) at urban and regional sites under Scenario 1. Blue bars represent the urban site, while yellow bars represent the regional site. Error bars (this study only) show

the range from using the maximum and minimum estimated photolysis rate constants in the simulations.

[Figure]

Figure S3 Diurnal variation of the ROx production rate from the photolysis of non-HCHO OVOCs at the Guangzhou urban site (a) and the PRD regional site (b) under Scenario 1. Red lines represent results from this study, while blue lines represent those from the MCM v3.3.1 mechanism. Shaded areas (this study only) show the range from using the maximum and minimum estimated photolysis rate constants in the simulations.

6. Figure 8 and Figure 9 (a, c, e) were not refered in the text.

Reply: Thank you for pointing this out. Figure 8 is already referenced at line 454. The missing reference to Figure 9 (a, c, e) has been added at line 488 in the revised manuscript.

7. The evaluation of the newly developed method could be further strengthened. As shown in Figure S5, the results from this study differ significantly from those estimated by previous approaches. Please include more explanation and discussion on this aspect in the manuscript.

Reply: Thank you for the helpful suggestion. The discrepancies observed in Figure S5 arise from the fact that the comparison data represent estimates from previous studies,

rather than directly measured data. These data come from eight different sources employing varied methodologies, including quantum chemical calculations, single-wavelength measurements, or derivations based on measured absorption cross-sections and assumed quantum yields. As such, the comparison data themselves carry substantial uncertainties, which is also why they were not used in constructing our parameterization. In addition, the current parameterization has not yet incorporated structure-specific adjustments for certain compound classes, such as monoterpene-derived organic nitrates, which may introduce additional bias in those cases. This has been clarified and discussed in the revised manuscript (lines 360-369):

*Notable deviations mainly originate from two studies. Wang et al. (2023) reported significantly lower photolysis rates for three classes of monoterpene-derived organic nitrates, while Liu et al. (2018) estimated the photolysis rates of α-hydroperoxycarbonyls to be only about 7% of our estimated values. These differences can be explained by the substantial uncertainties in the comparison dataset, which was derived from eight independent studies that employed a variety of methodologies, such as quantum chemical calculations, single-wavelength measurements, and indirect estimates based on absorption cross-sections combined with assumed quantum yields. In addition, the current method does not incorporate structure-specific adjustments for certain compound classes, which may also contribute to the observed discrepancies.*

**References**

Calvert, J. G., Mellouki, A., Orlando, J. J., Pilling, M. J., Wallington, T. J. (2011). Mechanisms of Photodecomposition of the Sunlight-Absorbing Oxygenates. In *Mechanisms of Atmospheric Oxidation of the Oxygenates* (pp. 974-1357): Oxford University Press.

Liu, Z., Nguyen, V. S., Harvey, J., Müller, J.-F., Peeters, J. (2018). The photolysis of α-hydroperoxycarbonyls. *Physical Chemistry Chemical Physics, 20*(10), 6970-6979.

Wang, Y., Takeuchi, M., Wang, S., Nizkorodov, S. A., France, S., Eris, G., et al. (2023). Photolysis of gas-phase atmospherically relevant monoterpene-derived organic nitrates. *The Journal of Physical Chemistry A, 127*(4), 987-999.

---

## Author Comment (AC2)

**Reviewer #2**

**General comments**

This study presents a novel and well-executed parameterization method for estimating the photolysis rate constants of oxygenated volatile organic compounds (OVOCs) based on molecular structure. By constructing a photolysis module that complements the MCM v3.3.1 mechanism, the authors successfully establish a structure–reactivity relationship that overcomes the longstanding limitation of insufficient quantum yield data. This allows for the estimation of photolysis rate constants for a wide range of compounds lacking experimental measurements. Furthermore, by integrating the updated photolysis mechanism, the study highlights the significant contribution of non-formaldehyde OVOCs to the daily average concentration of ROx radicals. Overall, this work introduces a creative and highly valuable approach for addressing a critical gap in atmospheric chemistry modeling. Some technical suggestions for improving the manuscript are provided below.

Reply: We sincerely thank the reviewer for the positive feedback and careful review. The reviewer's detailed suggestions and corrections were very helpful in improving the clarity and accuracy of our manuscript. Please find the response to individual comments below.

**Specific comments**

(1) Line 28, Please note that 'photolysis rate' is different from 'photolysis rate constant.' It would be more appropriate to use 'photolysis rate constant' here and in other relevant parts of the manuscript.

Reply: Thank you for pointing out this important terminology distinction. We have reviewed the manuscript and replaced "photolysis rate" with "photolysis rate constant" where appropriate, including in Line 28 and other relevant sections, to ensure scientific accuracy and consistency.

(2) Line 155, there is a Chinese-style period that needs to be replaced.

Reply: The Chinese-style period at line 155 (now at line 159) has been corrected and replaced with a standard English full stop. In addition, the manuscript has been carefully proofread to eliminate any remaining typographical errors.

(3)  The Supplementary Information includes numerous tables and figures; however, several figures are not referenced in the main text. It is recommended that the authors incorporate appropriate citations and briefly discuss the relevance of these figures to enhance clarity and coherence.

Reply: We have carefully reviewed the Supplementary Information and updated the main text to include explicit references to all relevant figures and tables. Brief descriptions of their significance have been added at appropriate locations in the manuscript or supplementary text to improve coherence and ensure that readers can better understand their context and relevance to the main discussion.